# Musculoskeletal Health in Active Ambulatory Men with Cerebral Palsy and the Impact of Vitamin D

**DOI:** 10.3390/nu13072481

**Published:** 2021-07-20

**Authors:** Christina Kate Langley, Gladys Leopoldine Onambélé-Pearson, David Thomas Sims, Ayser Hussain, Aidan John Buffey, Holly Leigh Bardwell, Christopher Ian Morse

**Affiliations:** 1Department of Sport and Exercise Sciences, Musculoskeletal Sciences and Sport Medicine Research Centre, Faculty of Science & Engineering, Manchester Metropolitan University, Manchester M15 6BH, UK; G.Pearson@mmu.ac.uk (G.L.O.-P.); d.sims@mmu.ac.uk (D.T.S.); ayser.hussain@thefa.com (A.H.); Aidan.Buffey@ul.ie (A.J.B.); hollybardwell@googlemail.co.uk (H.L.B.); c.morse@mmu.ac.uk (C.I.M.); 2The Football Association, St Georges Park, Burton-Upon-Trent DE13 9RN, UK; 3Department of Physical Education and Sport Sciences, University of Limerick, V94 T9PX Limerick, Ireland; 4Physical Activity for Health Cluster, Health Research Institute, University of Limerick, V94 T9PX Limerick, Ireland

**Keywords:** cerebral palsy, vitamin D, strength, bone, sun exposure

## Abstract

Purpose: (1) To determine the contribution of diet, time spent outdoors, and habitual physical activity (PA) on vitamin D status in men with cerebral palsy (CP) compared to physical activity matched controls (TDC) without neurological impairment; (2) to determine the role of vitamin D on musculoskeletal health, morphology, and function in men with CP compared to TDC. Materials and methods: A cross-sectional comparison study where 24 active, ambulant men with CP aged 21.0 ± 1.4 years (Gross Motor Function Classification Score (I–II) and 24 healthy TDC aged 25.3 ± 3.1 years completed in vivo assessment of musculoskeletal health, including: *vastus lateralis* anatomical cross-sectional area (VL ACSA), isometric knee extension maximal voluntary contraction (KE iMVC), 10 m sprint, vertical jumps (VJ), and radius and tibia bone ultrasound (US) T_us_ and Z_us_ scores. Assessments of vitamin D status through venous samples of serum 25-hydroxyvitamin D (25(OH)D) and parathyroid hormone, dietary vitamin D intake from food diary, and total sun exposure via questionnaire were also taken. Results: Men with CP had 40.5% weaker KE iMVC, 23.7% smaller VL ACSA, 22.2% lower VJ, 14.6% lower KE iMVC/VL ACSA ratio, 22.4% lower KE iMVC/body mass (BM) ratio, and 25.1% lower KE iMVC/lean body mass (LBM) ratio (all *p* < 0.05). Radius T_us_ and Z_us_ scores were 1.75 and 1.57 standard deviations lower than TDC, respectively (*p* < 0.05), whereas neither tibia T_us_ nor Z_us_ scores showed any difference compared to TDC (*p* > 0.05). The 25(OH)D was not different between groups, and 90.9% of men with CP and 91.7% of TDC had low 25(OH)D levels when compared to current UK recommendations. The 25(OH)D was positively associated with KE iMVC/LBM ratio in men with CP (r = 0.500, *p* = 0.020) but not in TDC (r = 0.281, *p* = 0.104). Conclusion: Musculoskeletal outcomes in men with CP were lower than TDC, and despite there being no difference in levels of 25(OH)D between the groups, 25 (OH)D was associated with strength (KE iMVC/LBM) in the CP group but not TDC. The findings suggest that vitamin D deficiency can accentuate some of the condition-specific impairments to musculoskeletal outcomes.

## 1. Introduction

Circulating vitamin D levels in vivo can account, in part, for observed variance of musculoskeletal outcome measures, with known implications for musculoskeletal impairments resulting from low vitamin D levels in otherwise healthy adults [1]. Predominant reasons for reduced vitamin D levels are due to insufficient dietary intake and restricted outdoor exposure to sun light [2]. Ultraviolet beta (UV b) radiation from sun exposure represents the primary source of endogenous vitamin D (serum 25-hydroxyvitamin D, 25(OH)D), and, as a result, endogenous vitamin D_3_ synthesis is severely decreased at latitudes over 35° N during winter months (the UK is ~53° N). Accordingly, individuals at these latitudes are susceptible to vitamin D deficiency [1] with 74.5% of the UK population exhibiting low vitamin D levels, comprising 33.7% who are vitamin D insufficient (25(OH)D levels of 30–20 ng mL^−1^) and 40.8% who are vitamin D deficient (25(OH)D < 20 ng mL^−1^).

The current recommendations for dietary intake of vitamin D for UK-based 19–50 year olds is 400–600 IU per day to benefit musculoskeletal function [3]. Within the general UK population, dietary intakes of vitamin D range from 168–291 IU/d (3.5–6.39 μg/d) in men and 140–255 IU/d (4.2–7.28 μg/d) in women, contributing to the high levels of the UK population who are vitamin D deficient [2]. Only a few foods, such as oily fish, naturally contain vitamin D, and these tend to be relatively low dosages [1], meaning it is difficult to meet these recommendations through diet alone. As approximately 80% of circulating 25(OH)D comes from UV b radiation from spending time outdoors in direct sunlight [1], it is important that individuals perform outdoor activities in order to accumulate vitamin D in replacement of the lack of endogenous vitamin D. Measures of both dietary vitamin D analysis [4] and outdoor sunlight exposure are heavily reported with typically developed, able-bodied adults and children (hereafter termed TDC, typically developed controls) [5,6]. Interestingly, individuals with disabilities, such as cerebral palsy and spinal cord injuries, appear to have poorer micronutrients compared to age-matched TDCs (Oleson, Patel [7,8]). Furthermore, although not directly related to outdoor activity, physical activity (PA) is 48 min lower, and sedentary behaviour is 80 min higher per day in men with CP with Gross Motor Function Classification Systems (GMFCS) of I–II compared to TDC (Nooijen, Slaman [9]), suggesting a lower level of UV b exposure in individuals with CP. Despite physically active individuals reporting higher sun exposure than more sedentary individuals, there are many studies that report low 25(OH)D in athletic populations with and without disabilities. For example, Morton, Iqbal [10] found that elite English Premier League footballers (living at a latitude of 53° N) had low vitamin D (mean ± standard deviation (SD), 20.5 ± 7.63 ng∙mL^−1^) during the winter months, while Flueck, Hartmann [11] showed 83.7% of 72 Swiss elite wheelchair athletes (living at a latitude of 47° N) were vitamin D insufficient.

The presence of low vitamin D levels in physically impaired groups, such as those with CP, is a particular concern given their diminished musculoskeletal health measures, such as muscle force production [12,13]. Lower levels of 25(OH)D are associated with lower musculoskeletal health outcomes including but not limited to: lower strength [14], muscle size [15], reduced bone mass [16], increased parathyroid hormone (PTH) [15], and higher levels of adiposity in TDC [17]. Individuals with CP are predisposed to reduced muscle size, strength, and bone mass, primarily due to reduced mechanical loading from altered gait patterns as a consequence of increased muscle tone and lower range of motion [18,19]. These already impaired musculoskeletal outcomes are compromised further in CP through modifiable risk factors including poor dietary micronutrient intake [20], reduced sun exposure [21], low PA levels [22], and as a side effect of anticonvulsant medications [23]. As described, the observed lower vitamin D levels in groups with CP could be a risk factor to exacerbated musculoskeletal health, but there appears to be no data pertaining to the measurements of vitamin D in PA groups with CP nor its effects on musculoskeletal measures.

Therefore, the aim of this present study was to increase our understanding of the role of vitamin D on musculoskeletal health in men with CP and compare it to TDC. The objectives of this study were to: (1) determine the impact of diet, time spent outdoors, PA, and vitamin D status in men with CP compared to TDC without neurological disabilities; (2) determine the role of vitamin D on musculoskeletal health in men with CP compared to TDC. The hypotheses of this study were: (1) men with CP have lower levels of vitamin D compared to TDC; (2) lower vitamin D leads to reduced musculoskeletal health in men with CP.

## 2. Materials and Methods

### 2.1. Study Protocol

The study took place in the UK at Manchester and Derby. A total of 48 volunteers participated in this study consisting of 24 ambulatory males with CP and 24 male TDC (Table 1). All participants provided written informed consent following approval from the local Ethics Committee. Participants were assessed for anthropometric measures, muscle size, bone ultrasound T_us_ and Z_us_ scores, muscle function, dietary vitamin D, total sun exposure, and PA (described below). All participants provided venous blood samples for subsequent total serum 25(OH)D and serum PTH analysis.

All participants were aged 18–30 years old and were UK residents having lived above 35° N for at least three months prior to this study. The CP group was ambulatory male footballers with cerebral palsy playing in the development or the elite disability football teams from The Football Association and playing football more than twice per week. Typically developed control participants were free of any neuromuscular disorder and pl6ayed football at least twice per week. Participants were excluded from the study if they: reported to have taken vitamin D supplements or used sun beds in the last 3 months prior to the study, went on regular holidays (defined as a destination between latitudes of 35° N and 35° S with a duration >7 days at a frequency >2 per year), had any illnesses (e.g., chronic kidney disease), or were known to be using any medication that may affect the metabolism of vitamin D (e.g., corticosteroids).

### 2.2. Participants and Recruitment

Men with CP (diplegic = 6, hemiplegic = 18) were recruited via The Football Association and were classified as FT3 (*n* = 3), FT2 (*n* = 17) and FT1 (*n* = 4) with GMFCS between I (*n* = 18) and II (*n* = 6, Table 1). All participants were tested on a single testing session with the same equipment. Testing commenced on 14 February 2019 and ended on 13 March 2019 for all groups, as total 25(OH)D levels were most likely to be near their nadir in the UK population based at a latitude of approximately 53° N [10,24].

#### 2.2.1. Anthropometric Measures

Height (m) was measured using a stadiometer (Seca 213, portable stadiometer, Hamburg, Germany) following the stretch-stature method [25] and body mass (BM) (kg) via a set of digital scales with minimal clothing (Seca, Hamburg, Germany). Percentages (%) of body fat and lean body mass (LBM) were measured using bioelectrical impedance (BIA) (Omron, body fat monitor, BF306, Kyoto, Japan). Participants stood upright with their arms out on front and gripped the electrodes on each handle. BIA was shown to be valid (R^2^ = 0.96) in comparison with dual energy X-ray absorptiometry in adult TDC [26] and children with CP (concordance correlation coefficient, 0.75–0.82) [27]. It should be noted that, where greater variance was reported between BIA and DEXA in CP populations, this was attributed in part to errors associated with the use of estimated standing height [27,28], whereas, in the present study, all CP participants were able to provide a standing height measure.

#### 2.2.2. Physical Activity

Habitual PA was recorded through the international physical activity questionnaire-long form (IPAQ) and presented as IPAQ score. The IPAQ consisted of 27 questions asking about the amount of time spent performing sedentary behaviours, light intensity physical activities, and moderate to vigorous physical activity around travel, work, and free time. In addition to PA questions, participants were also asked to answer questions on PA around occupation, transport, home, yard/garden, and leisure/sports. To assess habitual exercise, football training data were logged using 7 day diaries. Data collected included frequency of training (days per week), duration of each session (mins), and total time spent training (min per week). Step count was also recorded through mobile phone accelerometers from those participants (*n* = 46) with the iPhone Health Application (Apple Inc. Cupertino, CA, USA, version 13) as a daily average from the preceding 3 months.

#### 2.2.3. Muscle Size

Images of the *vastus lateralis* (VL) of the impaired leg of hemiplegic CP or the most paretic leg of those with diplegic CP and the dominant leg of TDC were obtained using B-mode ultrasonography with a 7.5 MHz linear array probe (MyLabGamma Portable Ultrasound, Esaote Biomedica, Genoa, Italy) to estimate the anatomical cross-sectional area (ACSA). As described by Reeves, Maganaris [29], the VL’s proximal insertion and the myotendinous junction were marked to identify 50% of muscle length. A strip of echo-absorptive markers spaced equally apart was placed horizontally around the VL to project a shadow onto the ultrasound image to provide a positional reference. With the probe in the transverse plane, a recording of the probe moving from the medial border on the VL to the lateral border of the VL was obtained. Individual images were extracted from the recording and used to construct the muscle by overlapping anatomical landmarks and external markers using Microsoft PowerPoint. ImageJ software (Version 1.41, National Institutes of Health, Bethesda, MD, USA) was used to measure the cross-sectional area of the constructed VL to determine VL ACSA [30]. Reeves, Maganaris [29] validated this technique against magnetic resonance imaging (MRI) and showed an inter class correlation (ICC) of 0.99 and mean typical error of 0.3 cm^2^. Throughout the experimental procedures, all unilateral measures were taken from the paretic side of hemiplegic participants and the dominant leg of TDC. In diplegic participants, their most affected side was measured. Due to time constraints of working with men with CP, it was impossible to measure both limbs, however, it should be noted that previous data showed that similar participants with CP showed no significant difference between their unaffected side and TDC [31].

#### 2.2.4. Muscle Function

To assess muscle function, vertical jump (VJ) height (m), maximum sprint time (s), grip strength (kg), and isometric knee extension maximal voluntary contraction (KE iMVC, N) were measured. Prior to the tests, all participants were taken through a standardised warm up which aimed to increase heart rate to over 120 bpm (Polar H10 chest heart rate monitor, Polar Electro, Kempele, Finland) and dynamic stretches with focus on the muscles around hips, knees, and ankles. Participants were given two attempts at each test, with 1 min rest in between, and the best result was recorded. VJ height (m) was measured using a jump mat (Probotics Inc., Esslinger court, Huntsville, Alabama) in two conditions—with and without arm swing. Nuzzo, Anning [32] reported the jump mat to be a reliable piece of equipment to measure VJ height in males (ICC = 0.93, coefficient of variation (CV) = 2.3%) and females (ICC = 0.90, CV = 6%) over two separate days.

Maximum sprint speed was assessed over 10 m. Two sets of sensory timing gates (Brower timing system, Wireless Sprint System 2007, Brower, Draper, UT, USA) were set up 1 m apart at either end of a 10 m distance. Participants performed two sprints with a standing start 0.60 m behind the first set of gates. This was shown to be a reliable method when measured on two separate days (ICC = 0.912, *p* < 0.01) [33].

Grip strength was assessed using a handgrip dynamometer (Jamar plus, Sammons Preston Rolyon, Bolingbrook, IL, USA). Participants chose their most comfortable grip position, and two maximal grip efforts were performed while standing with the elbow as extended as possible and the arm raised in front of the body level with the shoulder. Both tests were separated by 1 min, and the highest value was recorded. This current study showed a high test-retest reliability in men with CP and TDC (ICC = 0.996–0.998, both *p* < 0.001).

To record KE iMVC, participants were seated on a custom-made isokinetic chair fitted with a portable load cell (Manchester Metropolitan University, Manchester, UK). Their arms were across their chest, and the load cell attached around the dominant kicking leg (or the most paretic side in the CP group) with the knee at 90° flexion. The tested leg was fastened to a force transducer placed 5 cm above the lateral malleolus. Participants were instructed to extend the fastened leg maximally, and verbal encouragement was given during the measurement. Two trials were performed with 1 min break between each trial. The highest force produced was digitised using an analogue-to-digital converter, displayed by a self-displayed and coded program (MyLabView, National Instruments, Berkshire, UK) [34]. KE iMVC values were also presented relative to VL ACSA (KE iMVC/ACSA), BM (KE iMVC/BM), and LBM (KE iMVC/LBM).

#### 2.2.5. Bone Ultrasound

Bone ultrasound (US) was used to assess bone T_us_ and Z_us_ scores (Sunlight, BeamMed Ltd., Petah Tikva, Israel) of the distal radius (~5 cm from the condyle) and the distal tibia (~12 cm from the condyle). Participants laid supine for both measures. Ultrasound gel was applied to the skin surface at the measurement site to facilitate acoustic coupling. To assess the distal radius, the handheld probe was placed in the sagittal plane on the distal third of the radius. The probe was rotated ~70° laterally and ~70° distally in the horizontal axis around the radius slowly without lifting the probe from the skin surface. The distal third of the tibia was measured by placing the probe in the sagittal plane on the anterior portion of the tibia. The probe was moved back and forth ~4 cm in the transverse plane across the bone without uncoupling the probe from the skin surface. The measurements for each procedure were repeated 3–5 times depending on scan quality. After the signal was digitised and stored, the data were transferred to a computer for automated analysis, and a T_us_ and Z_us_ score was provided. Knapp, Blake [35] reported that Sunlight ultrasound systems have reliable intra-operator precision at distal radius: 0.36% (after 10 consecutive scans) and precise in vivo precision: 0.4–0.8% (scans were performed every 2 months for 2 years).

### 2.3. Dietary Vitamin D Assessment

To assess habitual dietary vitamin D intake, participants completed a 7 day food diary using a mobile phone application (Libro beta, Nutritics, Co. Dublin, Ireland). Participants logged the weight (g) of food used in all meals and snacks they consumed. This was then analysed in Nutritics Software, which provided dietary vitamin D in µg. Day, McKeown [36] found that food diary recall was a reliable method for micronutrient intake (R = 0.75) when compared to urinary markers.

#### Sun Exposure Measurement

To estimate the level of endogenous skin synthesis of vitamin D_3_ from sun exposure, a sun exposure questionnaire (SEQ) was used to assess frequency, time of day, and amount of time that they spent exposed to direct sunlight in spring and summer months [37]. Questions also included the type of sun protections that participants habitually used that were likely to inhibit vitamin D_3_ synthesis (i.e., SPF sun cream and clothing worn). To obtain a sunlight exposure score, a coded model was used based around the sun exposure questions and Fitzpatrick scale to give a total sun exposure (TSE) score for each participant [38].

### 2.4. Blood Sample Collection

Venous blood samples of 5 mL were taken from the antecubital region of the arm for all participants. Samples were collected via needle and eccentric luer tip syringe (Terumo corporation, Shibuya, Tokyo, Japan), transferred into vacutainer plain tubes (BD Vacutainer Plus^®^ plastic serum tube, Bristol Circle, Oakville, ON, Canada), and immediately centrifuged at 4500 G (Hermle, Model Z380, Countertop Centrifuge, Gosheim, Germany) for 10 min to separate the serum. Serum was removed via a micropipette calibrated to 100 µL (Pipetman pipette 10–100 µL, Gilson Scientific Ltd., Luton, UK) into two Eppendorf tubes (Eppendorf Tubes^®^ 3810X, Eppendorf, Hamburg) and stored at −20 °C.

#### Measurement of Serum 25(OH)D

Total 25(OH)D concentrations were measured using enzyme-linked immune-sorbent assay (ELISA) (Orgentec Diagnostika GmbH, Germany). The Orgentec ELISA showed a good correlation of R^2^ = 0.83 when compared to liquid chromatography mass spectrometry (LC-MS/MS) [39]. The manufacturer of the ELISA (Orgentec) provided intra- and inter-assay CV’s of <14.6% and <11.7%. The intra assay CV for 25(OH)D in our hands was in fact lower at 2%. A four-parameter logistic curve also showed a reliable calibration curve (optical density vs. concentration (ng·mL^−1^) R^2^ = 0.9819. The Orgentec ELISA was previously validated against the LC-MS/MS with a bias of 17.8% [40].

### 2.5. Measurement of Parathyroid Hormone

Serum PTH (PTH) was measured using a 90 min, one-wash ELISA (Abcam, Cambridge, UK). The ELISA had a range of 4.69–300 pg∙mL^−1^ with a sensitivity of 0.761 pg∙mL^−1^. The manufacturer of the ELISA (Abcam, Cambridge, UK) provided intra and inter-assay CVs of 1.5% and 3.8%, respectively; the intra-assay CV for PTH from this current study was 7.5%. A four-parameter polynomial curve also showed a strong reliability score of R^2^ = 0.9991 in this current study.

### 2.6. Statistical Analyses

Statistics were performed using SPSS statistics (SPSS Statistics 25, IBM Chicago, IL, USA). Data were assessed for normal distribution using a Shapiro–Wilks test (*p* > 0.05). Homogeneity of variance was assessed using Levene’s test, and a corrected *p* value was applied if variance was non-homogenous. Group differences for height, BM, BMI, BF%, LBM%, dietary intake, TSE, VJ no arms, VJ with arms, 10 m sprint, grip strength, KE iMVC/BM, KE iMVC/LBM, and KE iMVC/VL ACSA were assessed by independent T tests. Non-parametric group differences for age, tibia T_us_ and Z_us_ scores, radius T_us_ and Z_us_ scores, VL ACSA, and KE iMVC were assessed using Mann–Whitney U. Pearson correlations were performed to determine any relationships that exist between 25(OH)D and diet, UV b exposure, and bone T_us_ and Z_us_ scores. All data are presented as mean ± SD unless otherwise stated, and the confidence interval was set at 95% with alpha set at ≤0.05.

## 3. Results

### Population Comparisons for CP and TDC

The CP group was 4.3 years younger (*p* < 0.001) and had a 14.1% lower BM (*p =* 0.001), a 12.1% lower BMI (*p* < 0.001), a 20% lower BF% (*p* = 0.23), and a 20% higher LBM% (*p* = 0.023) compared to TDC (Table 2). Although not specifically matched for PA, there were no group differences in height, IPAQ, step count, PA frequency, PA average minutes per session, or PA total time per week between groups (*p* > 0.05, Table 2). Pearson’s correlations showed that age was not associated with any outcome measure (all *p* > 0.05).

The CP group had a 23.7% smaller VL ACSA (*p* = 0.001), and KE iMVC was 40.5% lower compared to TDC (*p* < 0.001, Table 3). KE iMVC/VL ACSA, KE iMVC/BM, and KE iMVC/LBM were 14.6% (*p* = 0.048), 22.4% (*p* = 0.004), and 25.1% (*p* = 0.002) lower, respectively, in CP compared to TDC (Table 3). The CP group had a 22.2% lower VJ with no arm swing and a 21.8% lower VJ with arm swing compared to TDC (*p* < 0.001 for both (Table 3). There were no differences in handgrip strength (*p* = 0.280) or 10 m sprint time between CP and TDC (*p* = 0.302, Table 3).

In both groups combined, there was a positive relationship between KE iMVC/LBM and jump height (no arms) (R = 0.368, *p* = 0.021) and VJ (with arm swing) (R = 0.351, *p* = 0.029), but KE iMVC/LBM was not associated with 10 m sprint time (R = 0.024, *p* = 0.881). There was, however, no relationship between KE iMVC/LBM and jump height (no arms), jump height (with arm swing), or 10 m sprint time in either men with CP (R = 0.232–0.379, *p* > 0.05) or TDC (R = 0.026–0.087, *p* > 0.05) when grouped separately.

The CP group had a radius T_us_ score that was −1.75 SDs less and a radius Z_us_ score that was −1.57 SDs less when compared to TDC (*p* < 0.001, Table 4). There was no difference between tibia T_us_ scores (*p* = 0.158, Table 4) and tibia Z_us_ scores between groups (*p* = 0.143, Table 4). A Pearson’s correlation showed that there were no significant relationships between 25(OH)D and any tibia or radius Z_us_ and T_us_ scores for both groups (R = 0.162–0.253, all *p* > 0.05). There were also no significant relationships between PTH and any tibia or radius Z_us_ and T_us_ scores for both groups (R = 0.012–0.205, all *p* > 0.05).

There were no differences in 25(OH)D ng∙mL^−1^ (*p* = 0.381), PTH (*p* = 0.710), dietary intake (*p* = 0.540), or TSE score between groups (*p* = 0.790, Table 5). Of the men with CP, 5/22 (22.7%) were classed as severely deficient, 7/22 (31.8%) were deficient, 8/22 (36.4%) were insufficient, and 2/22 (9.1%) were adequate in 25(OH)D, while 8/24 (33.3%) of the TDC were classed as severely deficient, 9/24 (37.5%) were deficient, 5/24 (20.8%) were insufficient, and 2/24 (8.3%) were adequate in 25(OH)D. Pearson’s correlation showed no relationship between 25(OH)D and dietary vitamin D in men with CP (R = 0.079, *p* = 0.798) or TDC (R = 0.165, *p* = 0.607, Table 5), nor was there a relationship between 25(OH)D and TSE in men with CP (R = 0.041, *p* = 0.857) or TDC (R = 0.167, *p* = 0.447, Table 5). Pearson’s correlation showed that 25(OH)D levels were associated with stronger KE iMVC/LBM (R = 0.500, *p =* 0.020 (1-tailed)) in men with CP (Figure 1), but there was no association between 25(OH)D and KE iMVC/LBM in TDC (R = 0.281, *p* = 0.103 (1-tailed), Figure 1).

## 4. Discussion

The aim of this study was to increase the understating of vitamin D and musculoskeletal health in men with CP. Our findings show that (1) the CP participants had lower KE strength, smaller VL ACSA, and lower VJ and sprint performance compared to TDC; (2) bone T_us_ and Z_us_ group comparisons were site-specific to upper and lower limbs, with the CP group showing lower radius scores but no difference in the tibia from TDC; (3) 25(OH)D levels were similar between CP and TDC groups, however, almost all participants were below levels considered sufficient; and (4) there was a positive association between 25(OH)D and KE iMVC/LBM in the CP group but no association in the TDC group.

Due to previous associations between impaired walking and plantar flexors (PF) weakness in adults with CP [41], most studies in CP measured PF strength and PF ACSA [12,42,43]. However, the KE was deemed an important muscle group to investigate, as these muscles play a dominant role in sports performance [44] and fall prevention [45] and are limited in their description within CP [46]. The 40.5% lower KE iMVC in the present study was consistent with the 52% weaker PF [47] and was a consequence of the PF being more directly impacted by the CP condition [12]. In contrast to our more modest levels of KE weakness, the previous report of a 69% weaker KE in CP [46] likely reflects the sex difference in strength between their mixed sex CP participants compared to their male only TDC. In the present study, men with CP had 23.7% smaller VL ACSA, which was consistent with the 20% smaller PF ACSA and the 27.9% smaller VL ACSA in men with CP compared to TDC (Noble, Fry [42]). The men with CP in this current study had a higher LBM% of 86.5% compared to other studies which used BIA in young ambulatory males with CP who had an LBM percentage of 74.8% [28]; this could be attributed to the high levels of PA in the current men with CP equivalent to 3996 steps more/day when compared to other young, ambulatory CP participants [48].

A lower knee extensor strength relative to LBM has a number of functional implications in both CP and other conditions of muscle weakness. Beyond our population of men with CP, a lower KEMVC/BM is an observation that was made in a number of conditions such as: obese aging men [49], postmenopausal women [50], and individuals with degenerative muscle impairments [51]. Due to the non-progressive nature of CP on neuromuscular properties, strategies to train and improve the quality (contractile properties, size, and strength) of the effected muscles, particularly the lower limbs, need to be considered alongside body fat management to contribute to improved neuromuscular function.

Surprisingly, there was no difference in 10 m sprint time between groups in the current study. A combination of factors may account for the lack of 10 m sprint difference. One could be attributed to 18/24 (75%) of the current CP group classifying as GMFCS I. In higher GMFC scoring adults with CP, there is increased gait symmetry to TDC [52], which could contribute to effective ground reaction force vectors when sprinting [53]. de Groot et al. [46] also found no difference in peak power output in a CP group with a GMFCS of I when compared to TDC. It should also be considered that the majority of men with CP were hemiplegic in this study (Table 1), and running gait patterns were shown to have increased symmetry with increasing running speeds [54]. Therefore, peak power output and increased sprinting gait symmetry may explain why there was no difference in the 10 m sprint.

The bone T_us_ and Z_us_ score group differences showed site dependence with a lower radius T score of −1.32 representing a bone fracture risk of 2.3-fold greater compared to normative values [55]; in contrast, the distal tibia showed no difference between groups. This was similar to bone T_us_ and Z_us_ score data presented previously from adolescents with CP who had radius and tibia T scores of −1.07 and −0.38, respectively [56]. In adolescents with CP, tibia density was previously reported to be higher with increasing ambulation levels [57]. Indeed, football is specifically effective at preserving age related bone T_us_ and Z_us_ scores in those without CP [58]. It is therefore likely that our observations of preserved tibia T_us_ and Z_us_ scores may be a consequence of regular lower limb bone loading through activities such as football and associated exercise. For future research, it is important to acknowledge that the present study does not show that football alone improves bone T_us_ and Z_us_ scores in men with CP, and more work is required to show this association with the sport. It is more probable that the general activity levels of the CP group, which included large elements of football, matched T_us_ and Z_us_ scores of their tibia to TDC. Certainly though, clinicians and physical trainers should incorporate exercises that load the upper limbs to ensure bone turnover and development occurs in groups with CP.

The lack of difference in PTH between groups was consistent with the fact that almost all participants had low levels 25(OH)D [59]. There is a paucity of data on vitamin D concentration in disabled populations, particularly in regard to athletes with disabilities, despite low vitamin D concentrations being documented in several nonathletic populations [14,15,16]. Despite our findings that there are no differences in 25(OH)D between the men with CP and TDC, both groups were, on average, classified as deficient, with an insufficiency prevalence of ~90%. These low vitamin D levels relative to summer values are the norm in the UK during the winter months [60]. Yet, in the present study, low vitamin D levels may exacerbate the condition-specific weakness, e.g., 25(OH)D explained 26% of the variance in KE iMVC/LBM in the CP participants. In contrast, we saw no associations between 25(OH)D and other outcomes, a likely consequence of the very low levels of 25(OH)D. This was particularly noticeable in the lack of association between 25(OH)D and either dietary vitamin D or TSE in both participant groups, likely as all three could be considered very low, e.g., the dietary vitamin D intake (185.5 ± 155 IU/d) was not even close to the recommended value of 400–600 IU/d in adults [3]. Similarly, that there was no association between TSE and 25(OH)D in this current study was consistent with negligible UV b radiation from sun exposure during the latter winter months in the UK (i.e., February–March). Thus, even with the highest TSE scores, 25(OH)D would likely not have been high [60]. Despite the negligible 25(OH)D variance between groups, the role of 25(OH)D in KE MVC/BM was consistent with the well-established role of 25(OH)D on skeletal muscle myogenesis, cell proliferation, differentiation, regulation of protein synthesis, and mitochondrial metabolism [61]. With this knowledge (and the prevalent insufficiency in all participants), interventions such as vitamin D supplementation should be sought by men with both CP and TDC to correct for low vitamin D. In the future, it is important that seasonal variations in vitamin D are measured to identify if increased UV b radiation from sun exposure improves 25(OH)D to adequate levels and if musculoskeletal health outcome measures are impacted in men with CP and other para-athletes.

## 5. Strengths and Limitations

The participants in the present study represent a highly functional proportion of men with CP (GMFCS I-II). Our participant groups were, however, similar to the only other description of KE iMVC and sprint outcomes [46] and were consistent with populations of CP athletes investigated by others [62]. Comparisons with population-based studies suggest that the proportion of GMFCS I:GMFCSII participants is around *n* = 3:2 [63]. In contrast, the strength comparisons made in the present study as well as the studies by de Groot, Dallmeijer [46], and [62] were made in ambulatory participants with CP at a ratio of GMFCS I:GMFCS II, *n* = 3:1, with more severe impairments also included in O’Brien, Carroll [62]. The high levels of PA in these active CP participants, however, mean that the findings may not be generalisable to other more impaired populations with CP. Although a broader population would, of course, be relevant for generalisation, our population of lower GMFCS impairment likely reduces the contribution of PA variance to the group differences and suggests that, by regularly undertaking football and other forms of PA, there is some benefit to the musculoskeletal health of men with CP, particularly in the bone T_us_ and Z_us_ scores of the lower limbs. As addressed throughout these limitations, and consistent with the caveat of all studies, the outcomes reflect the population under investigation. To this end, it is important to note we include no data from women. Although not adverse to presenting sex disaggregated musculoskeletal outcomes such as KE strength and tendon stiffness [64,65], our recruitment of CP participants was driven by the prevailing opportunities for those with CP and, noticeably, a much more limited development pathway for women’s para-sport. In previous studies including active CP participants where women were presented, it was as a minority (men: women, 2:1 and 4:1, respectively [46,66]. Given the known sex differences in bone measures and the greater risk for osteoporosis [67], future research should strive to uncover particular risks and implications of low vitamin D in women with CP.

## 6. Conclusions

The aim of this study was to increase the understating of vitamin D and musculoskeletal health in men with CP. It illustrated that men with CP have lower KE iMVC force, smaller VL ACSA, reduced muscle function, and site-specific reduced T_us_ and Z_us_ scores of the radius compared to TDC. Men with CP and TDC are both also at high risk of low vitamin D during the winter months, which could contribute to weakness of the KE iMVC/LBM in men with CP. Therefore, it is important that men with CP undertake strategies to amend low vitamin D during the winter months such as vitamin D supplementation to ensure that decreased muscular performance and potential risk of falls from exacerbated knee extensor weakness are reduced.

## Figures and Tables

**Figure 1 nutrients-13-02481-f001:**
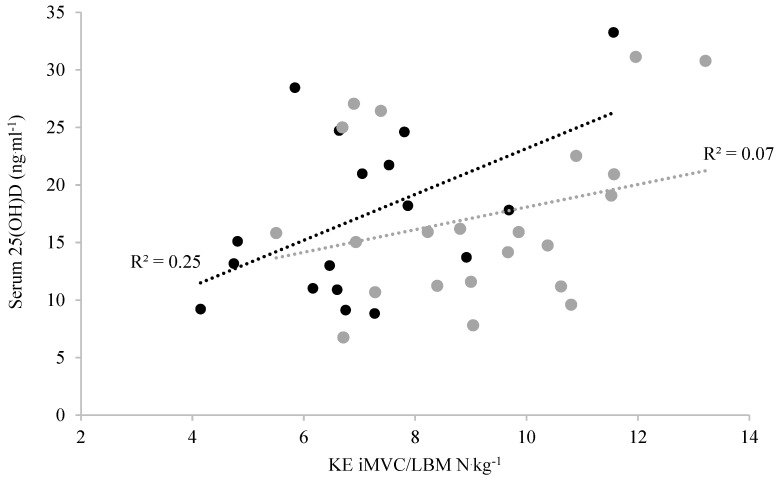
Pearson correlations between serum 25(OH)D and KE iMVC/LBM in men with CP (black filled dots) and TDC (grey filled dots).

**Table 1 nutrients-13-02481-t001:** Classification and impairment details of participants.

		CP Diplegic *n* = 6	CP Hemiplegic *n* = 18	CP Total *n* = 24	TDC *n* = 24
GMFCS	I	-	18	18	-
	II	6	-	6	-
IFCPF classification (FT)	1	4	-	4	-
	2	2	15	17	-
	3	-	3	3	-
Side measured	Left	5	7	12	4
	Right	1	11	12	20

GMFCS, Gross Motor Function Classification Score; IFCPF, International Federation of Cerebral Palsy Football.

**Table 2 nutrients-13-02481-t002:** Anthropometric and physical activity data in men with cerebral palsy (CP) and PA matched controls (TDC).

	CP	TDC	*p*
Age (y)	21.0 ± 1.4	25.3 ± 3.1	<0.001
Height (m)	1.74 ± 0.07	1.76 ± 0.08	0.335
BM (kg)	66.4 ± 10.1	76.5 ± 10.2	0.001
LBM (kg)	57.5 ± 9.8	64.0 ± 9.3	0.025
BMI (kg∙m^−2^)	21.9 ± 2.1	24.7 ± 2.3	<0.001
BF%	13.5 ± 4.4	16.5 ± 4.4	0.023
LBM%	86.5 ± 4.4	83.5 ± 4.4	0.023
GMFCS (mean (range))	1.2 (1–2)	-	-
IFCPF classification (mean (range))	2 (1–3)	-	-
IPAQ score	8384 ± 4720	7178 ± 5626	0.484
PA frequency (days/week)	4.00 ± 1.84	4.33 ± 1.86	0.535
PA duration (mins/session)	65.2 ± 28.3	77.9 ± 17.4	0.066
PA total time (mins/week)	251.3 ± 135.0	320.2 ± 149.5	0.100
Step count (steps/day)	8218 ± 3292	6943 ± 2295	0.126

Reported as mean ± SD and *p* values; BMI, BM index; BF%, body fat percentage; LBM, lean body mass; GMFCS, Gross Motor Function Classification Score; IFCPF, International Federation of Cerebral Palsy Football; IPAQ, international physical activity questionnaire; PA, physical activity.

**Table 3 nutrients-13-02481-t003:** Neuromuscular outcome measures in men with cerebral palsy (CP) and PA matched controls (TDC).

	CP	TDC	*p*
VL ACSA (cm^2^)	27.1 ± 5.4	34.4 ± 8.3	0.001
KE iMVC (N)	398.8 ± 94.3	601.4 ± 152.4	<0.001
KE iMVC/VL ACSA (N∙cm^−2^)	15.2 ± 3.13	17.6 ± 4.15	0.048
KE iMVC/BM (N∙kg^−1^)	6.07 ± 1.41	7.60 ± 1.65	0.004
KE iMVC/LBM (N∙kg^−1^)	7.11 ± 1.80	9.15 ± 2.06	0.002
VJ no arms (m)	0.40 ± 0.04	0.50 ± 0.05	<0.001
VJ with arms (m)	0.45 ± 0.04	0.56 ± 0.05	<0.001
Grip strength (kg)	39.8 ± 11.9	45.6 ± 7.4	0.280
10 m sprint (s)	1.90 ± 0.14	1.86 ± 0.12	0.302

Reported as mean ± SD and *p* values; VL ACSA, vastus lateralis anatomical cross-sectional area; KE iMVC, knee extensor isometric maximal voluntary contraction; BM, BM; VJ, vertical jump.

**Table 4 nutrients-13-02481-t004:** Tibia and radius bone ultrasound (Tus and Zus score) in men with cerebral palsy (CP) and PA matched controls (TDC).

	CP	TDC	*p*
Radius T_us_ score	−1.32 ± 1.12	0.43 ± 0.79	<0.001
Radius Z_us_ score	−0.93 ± 1.12	0.64 ± 0.79	<0.001
Tibia T_us_ score	0.50 ± 1.62	−0.03 ± 0.80	0.158
Tibia Z_us_ score	0.55 ± 1.60	−0.03 ± 0.82	0.143

Reported as mean ± SD and *p* values.

**Table 5 nutrients-13-02481-t005:** Vitamin D outcome measures in men with cerebral palsy (CP) and PA matched controls (TDC).

	CP	TDC	*p*
25(OH)D (ng·mL^−1^)	18.7 ± 7.3	16.9 ± 7.1	0.381
PTH (pg·mL^−1^)	25.1 ± 10.2	31.8 ± 14.2	0.710
Dietary intake (IU·day^−1^)	166 ± 186	205 ± 124	0.540
TSE score	27.4 ± 2.4	28.6 ± 2.1	0.790

Reported as mean ± SD and *p* values; 25(OH)D, 25-hydroxyvitamin D; PTH, parathyroid hormone; TSE, total sun exposure.

## Data Availability

Data can be made available on request.

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
