# Peer review of "Musculoskeletal Health in Active Ambulatory Men with Cerebral Palsy and the Impact of Vitamin D"

_nutrients, 2021, doi:10.3390/nu13072481_

Round 1

Reviewer 1 Report

Langley et al. report on vitamin D levels in cerebral palsy patients and compare their physical activities and performance with healthy age-matched control persons. While it is important to inform about low vitamin D levels in physically impaired groups, their conclusions are not supported by their results.

Major comments:

  • The suggestion of “a greater sensitivity to low vitamin D in men with CP with regards to bone and muscle content and functional outcomes” is not supported by the study data. How and why should this be proven without any molecular biological approaches? The results might only support different muscle activity at best – but this needs also other technologies.
  • The study ignores the influence of especially the NEUROmuscular local effects of the nerves on bone and muscle while looking only at systemic vitamin D levels.
  • In the study aims (line 106): why was the second hypothesis used? “Vitamin D will led to reduced musculoskeletal health” is contradictory to everything that was said in the introduction before.
  • Measuring vitamin D by ELISA with a very high coefficient of variation is not accepted in most of the respective journals, unless verified with a MS method.
  • The age of the groups was statistically different (at least 4 years!), which might greatly contribute to the results described. Other important parameters were also highly significantly different (BMI etc.) – this is not an “age-matched” setting.
  • “Bone health” cannot be described by bone ultrasound only. Writing about T- and Z-scores without mentioning of the device used (ultrasound) is not accepted, the authors might compare the usual literature sources. Ultrasound measurements might present some material properties, but they cannot be named as “bone densitometry” such as DXA. So, the term “bone health group differences” is not allowed.
  • The discussion is way too long and speculative. The simple conclusion, vitamin D should be supplemented, when necessary, is important, but not supported by most of the data and the design.

Minor comments:

  • Line 92: PTH is not a bone turnover marker, but a calciotropic hormone.
  • Exclusion criterion “using sun beds more than once per week” is per se not a good idea, implying the participants had access to a lot of skin vitamin D production. Vice versa, there is no exclusion of vitamin D supplements, at least in the text (!).

Reviewer 2 Report

For the purposes to increase the understanding of vitamin D and musculoskeletal health in men with CP (cerebral palsy), this manuscript investigates the vitamin D status and the role of vitamin D on musculoskeletal health and function in men with CP. The authors found that men with CP and age-matched controls (TDC) without neurological impairment both have a high risk of low vitamin D during the winter months. Men with CP have lower KE iMVC force, smaller VL ACSA, reduced muscle function, and site-specific reduced bone health of the radius than TDC. Concerning bone and muscle content and functional outcomes, this manuscript suggested that men with CP have a greater sensitivity to low vitamin D. The manuscript is easy to understand, and the data will help encourage men with CP to supplement vitamin D and to increase sun exposure times during the winter months to ensure a reduction in muscle performance and the potential risk of falls from extensor knee extensor weakness are reduced.

No comment.

Reviewer 3 Report

The article touches the issue of the significance of the role of vitamin D for musculoskeletal health in men with CP.

Comments to the authors.

The research includes too small group of participants to generalize the conclusions.

Line 106, the second hypothesis is incorrect, probably the authors wanted to refer to the insufficient level of vitamin D.

line 186 -7  "Their arms were across their chest and the load cell attached around the

dominant kicking leg (or the most paretic side in the CP group) with their knee at 90° flexion" -please explain -when the authors chose the dominant and when the most paretic leg.

line 416: BIA - no explanation what the abbreviation stands for
